# The Nucleoside/Nucleotide Analogs Tenofovir and Emtricitabine Are Inactive against SARS-CoV-2

**DOI:** 10.3390/molecules27134212

**Published:** 2022-06-30

**Authors:** Joy Y. Feng, Venice Du Pont, Darius Babusis, Calvin J. Gordon, Egor P. Tchesnokov, Jason K. Perry, Vincent Duong, Arya Vijjapurapu, Xiaofeng Zhao, Julie Chan, Cal Cohen, Kavita Juneja, Tomas Cihlar, Matthias Götte, John P. Bilello

**Affiliations:** 1Gilead Sciences, Inc., Foster City, CA 94404, USA; venice.dupont@gilead.com (V.D.P.); darius.babusis@gilead.com (D.B.); jason.perry@gilead.com (J.K.P.); vincent.duong1@gilead.com (V.D.); arya.vijjapurapu@gilead.com (A.V.); xiaofeng.zhao@gilead.com (X.Z.); julie.chan@gilead.com (J.C.); cal.cohen@gilead.com (C.C.); kavjuneja@gmail.com (K.J.); tomas.cihlar@gilead.com (T.C.); john.bilello@gilead.com (J.P.B.); 2Department of Medical Microbiology and Immunology, University of Alberta, Edmonton, AB T6G 2E1, Canada; cjgordon@ualberta.ca (C.J.G.); tchesnok@ualberta.ca (E.P.T.); gotte@ualberta.ca (M.G.)

**Keywords:** SARS-CoV-2, HIV-1 NRTI, tenofovir, TAF, TDF, FTC

## Abstract

The urgent response to the COVID-19 pandemic required accelerated evaluation of many approved drugs as potential antiviral agents against the causative pathogen, severe acute respiratory syndrome coronavirus 2 (SARS-CoV-2). Using cell-based, biochemical, and modeling approaches, we studied the approved HIV-1 nucleoside/tide reverse transcriptase inhibitors (NRTIs) tenofovir (TFV) and emtricitabine (FTC), as well as prodrugs tenofovir alafenamide (TAF) and tenofovir disoproxilfumarate (TDF) for their antiviral effect against SARS-CoV-2. A comprehensive set of in vitro data indicates that TFV, TAF, TDF, and FTC are inactive against SARS-CoV-2. None of the NRTIs showed antiviral activity in SARS-CoV-2 infected A549-hACE2 cells or in primary normal human lung bronchial epithelial (NHBE) cells at concentrations up to 50 µM TAF, TDF, FTC, or 500 µM TFV. These results are corroborated by the low incorporation efficiency of respective NTP analogs by the SARS-CoV-2 RNA-dependent-RNA polymerase (RdRp), and lack of the RdRp inhibition. Structural modeling further demonstrated poor fitting of these NRTI active metabolites at the SARS-CoV-2 RdRp active site. Our data indicate that the HIV-1 NRTIs are unlikely direct-antivirals against SARS-CoV-2, and clinicians and researchers should exercise caution when exploring ideas of using these and other NRTIs to treat or prevent COVID-19.

## 1. Introduction

Coronavirus-related outbreaks have been documented back to 1200–1500, and seen as recently as the SARS-CoV in 2002 and MERS-CoV in 2012 [1]; however, none of the prior encounters prepared us for the fast onset and spread of the COVID-19 pandemic of the last two years. To address the urgent COVID-19 medical need, evaluation of approved drugs as potential antiviral agents was conducted at an accelerated pace not seen since the identification of HIV as the cause of AIDS. In the past four decades, nucleoside and nucleotide analogs have served as cornerstones for the treatment of viral infections including herpes simplex virus (HSV), HIV-1, hepatitis B virus (HBV), and hepatitis C virus (HCV). Structurally, these analogs belong to two major groups: 2′-deoxy nucleoside/tides (AZT, d4T, ddI, TFV prodrugs, and FTC) for DNA viruses and retroviruses such as HSV, HIV-1, and HBV, and 2′-OH nucleotides (sofosbuvir) for RNA viruses such as HCV.

In 2020–2021, some groups suggested approved nucleoside/tide reverse transcriptase inhibitors (NRTIs), such as tenofovir (TFV) and emtricitabine (FTC) (Figure 1), as potential COVID-19 therapeutic candidates based on in silico molecular docking analysis [2,3] and SARS-CoV-2 RdRp biochemical assays [4,5]. In 2020, Boulle et al. conducted a risk factor analysis on COVID-19 death using data from a total of 3,460,932 adults attending public-sector health facilities in the Western Cape Province, South Africa. The authors reported that among COVID-19 cases in people living with HIV-1 (PLWH) on antiretroviral therapy (ART), a reduced COVID-19 mortality was observed in people receiving tenofovir disoproxil fumarate (TDF) versus other therapies. [6]. However, this notion is challenged by Ssetongo et al. (2021) in a systematic review and meta-analysis on the epidemiology and outcomes of COVID-19 in HIV-infected individuals [7], where the authors concluded that the beneficial effects of tenofovir in reducing risks of SARS-CoV-2 infection and death from COVID-19 in PLWH was inconclusive. At the end of 2021, Del Amo et al. reported that TDF/FTC lowers COVID-19 severity among HIV-positive individuals with HIV virologic control, based on observation studies from 51,558 eligible individuals from 69 clinics in Spain (medRxiv preprint, https://doi.org/10.1101/2021.11.11.21266189 accessed on 1 June 2022). To address the discordant reports on whether TFV and FTC are direct-acting SARS-CoV-2 antivirals, we studied TFV and FTC, as well as the prodrugs tenofovir afenamide (TAF) and TDF (Figure 1) with four approaches: (1) cell-based antiviral assays in A549-hACE2 cells and primary normal human bronchial epithelial (NHBE) cells; (2) assessment of the active metabolites TFV-DP and FTC-TP (Figure 1) incorporation efficiency into RNA by the SARS-CoV-2 RNA-dependent-RNA polymerase (RdRp); (3) evaluation of TFV-DP and FTC-TP inhibition of SARS-CoV-2 RdRp RNA synthesis; (4) structural modelling of the pre-incorporated active metabolites into the SARS-CoV-2 RdRp and a comparison to existing HIV-1 reverse transcriptase (RT) X-ray structures. In these studies, the approved direct-acting SARS-CoV-2 antiviral remdesivir (RDV) and its active triphosphate metabolite were used as positive controls. None of the HIV-1 NRTIs tested were active against SARS-CoV-2 in cell-based and enzymatic assays, supported by their poor fitting at the SARS-CoV-2 RdRp active site. Our findings demonstrated that a 2′-OH is needed for an NTP analog to be recognized as a substrate by the SARS-CoV-2 RdRp, and further support the time-tested principle in nucleoside/tide in antiviral drug discovery that viral RNA synthesis inhibitors are structurally distinct from viral DNA synthesis inhibitors.

## 2. Results

### 2.1. TAF, TDF, TFV, and FTC Are Inactive against SARS-CoV-2 in Cell-Based Assays

The antiviral effects of TAF, TDF, TFV, and FTC against SARS-CoV-2 were evaluated using SARS-CoV-2-Nluc and SARS-CoV-2-Fluc recombinant reporter viruses. All four compounds exhibited EC_50_ values >50 μM, whereas the RDV positive control demonstrated clear antiviral effect with an EC_50_ value of 0.104 ± 0.016 µM. Similarly, TAF, TDF, and FTC exhibited >50 μM EC_50_ values in NHBE antiviral assays, while RDV had an EC_50_ value of 0.037 ± 0.017 μM (Table 1). In comparison to the clinically relevant exposures of TAF, TDF, and FTC, the concentrations used in the cell-based assays are >10 or >100-fold higher, further suggesting it is highly unlikely for these NRTIs to achieve direct antiviral effects under the approved dosing regimen (Table 1).

### 2.2. Efficient Formation of Active Metabolites Are Observed in A549 Cells

The intracellular metabolism of FTC, TAF, TDF, TFV, and RDV was assessed in A549 cells following incubation with each compound to determine activation to and quantitative levels of their respective active diphospho-phosphonate (DP) or triphosphate metabolites (TP). Consistent with results previously reported in noted target cells (e.g., FTC [10] and TDF/TAF [11] in PBMC; RDV [12] in NHBE), efficient activation to respective active metabolite was also observed with all compounds in A549 cells. Using an intracellular volume of 1.67 pL/cell, at 24-h post-compound addition, intracellular concentrations of 21.0 ± 2.2 µM (RDV-TP from 1 µM RDV), 126 ± 50 (TFV-DP from 1 µM TAF), 14 ± 2 (TFV-DP from 1 µM TDF), 1.7 ± 0.6 µM (TFV-DP from 10 µM TFV), and 5.8 ± 1.8 µM (FTC-TP from 1 µM FTC) were achieved (Table 2).

### 2.3. TFV-DP and FTC-TP Are Poor Substrates of SARS-CoV-2 RdRp

It was previously shown that the RdRp of SARS-CoV-2 incorporated the pharmacologically active RDV triphosphate (RDV-TP) species approximately three-fold more efficiently than its natural counterpart ATP [14]. Moreover, the biochemical evaluation of viral RdRp from different RNA viruses demonstrated efficient RDV-TP incorporation is required to elicit an antiviral effect [15]. To address the differences observed in the antiviral activity of TFV-DP and FTC-TP against SARS-CoV-2, we determined the efficiency at which the NRTIs were incorporated by SARS-CoV-2 RdRp (Table 3). Short primer/templates mimicking a random elongation complex were employed to monitor SARS-CoV-2 RdRp-catalyzed RNA synthesis (Figure 2A). Under steady-state conditions, the incorporation efficiency of TFV-DP and FTC-TP were compared to ATP and CTP, respectively. Efficiency of each analog or natural counterpart was determined by monitoring a single incorporation at a fixed time point across a range of concentrations (Figure 2B). In the absence of a competing natural counterpart, TFV-DP and FTC-TP served as poor substrates for SARS-CoV-2 RdRp and the signal generated at position six as a result of NRTI incorporation was below the level of quantification (Figure 2C,D). Therefore, it is very unlikely that these nucleotide analogs would be incorporated into the nascent viral RNA strand synthesized by SARS-CoV-2 RdRp. The inability of SARS-CoV-2 RdRp to utilize TFV-DP and FTC-TP as a substrate explains the poor antiviral effect observed in cell culture (Table 1). To ensure the TFV-DP and FTC-TP were active, their incorporation by HIV-1 RT was monitored at a single position via a radiolabelled primer (Figure 3A). In similar fashion as above, incorporation of TFV-DP, FTC-TP, and their respective natural counterparts was visualized across titrated concentrations (Figure 3B). Quantification revealed TFV-DP and FTC-TP were incorporated 1.5- and 2-fold less efficiently than their natural counterparts, respectively (Figure 3C,D).

### 2.4. TFV-DP and FTC-TP Do Not Inhibit RNA Synthesis by SARS-CoV-2 RdRp

We compared the inhibitory effects of RDV-TP, TFV-DP, FTC-TP on SARS-CoV-2 RdRp-catalyzed RNA synthesis. Although we demonstrated that TFV-DP and FTC-TP are poor substrates for SARS-CoV-2 RdRp, we hypothesized that the nucleotide inhibitors could potentially compete with natural nucleotides without being incorporated into a growing strand. To test this possibility, we measured SARS-CoV-2 RdRp activity using a gel-based and radiometric filter binding assay in the presence of titrated amounts of TFV-DP or FTC-TP. For the gel-based assay, a random elongation complex was mimicked by employing an RNA primer/template substrate supplemented with a nucleotide cocktail mix. The 14-mer RNA template contained a single incorporation site for each respective nucleotide analogue at position six (Figure 4A). RDV-TP incorporation resulted in the formation of an intermediate product three nucleotides downstream of incorporation, which increased in a dose-dependent manner. RDV-TP inhibition distal from the site of incorporation is often defined as “delayed-chain termination” and is consistent with previous publications [14,15,16,17]. Under competitive conditions (i.e., both the natural nucleotide and the analog are present in the reaction mixture albeit at varied ratios), both TFV-DP and FTC-TP did not demonstrate inhibitory effects on SARS-CoV-2 RdRp-catalyzed RNA synthesis (Figure 4B). The RNA synthesis pattern described above correlated with the findings from the radiometric filter binding assay in which we measured inhibition of SARS-CoV-2 RdRp by RDV-TP as a positive control and obtained an IC_50_ of 1.0 μM (Table 4; Figure 4C,D). Compared to RDV-TP, neither TFV-DP nor FTC-TP inhibited RNA synthesis by SARS-CoV-2 RdRp using concentrations up to 10 μM. These results indicate the NRTIs are unable to compete with natural nucleotides and therefore do not elicit an inhibitory effect on SARS-CoV-2 RdRp-catalyzed RNA synthesis.

### 2.5. TFV-DP and FTC-TP Do Not Fit Well into the SARS-CoV-2 RdRp Active Site

The SARS-CoV-2 RdRp active site specifically selects for ribose nucleoside triphosphate substrates over 2′-deoxyribose substrates through a set of polar residues which recognize the ribose 2′OH. As shown in Figure 5A for the natural substrate, ATP, these residues include D623, S682 and N691, which specifically interact with the pre-incorporated nucleotide’s 2′OH. These particular residues are conserved in the viral RdRp’s of HCV [18], poliovirus [19], and norovirus [20], among others. Additional residues, T680 and T687, also contribute to the polar nature of the 2′ pocket in SARS-CoV-2. The 2′-OH of RDV-TP is similarly recognized by these residues, while its 1′-CN also fits particularly well in a partially overlapping polar pocket comprised of T687, N691, and S759, giving it an advantage for incorporation over ATP. Neither TFV-DP (Figure 5B) nor FTC-TP (Figure 5C) have a comparable 2′-OH, and instead present a hydrophobic face to this polar pocket. In contrast, HIV-1 RT selects for deoxyribose nucleoside triphosphate substrates (Figure 5D) by providing a hydrophobic environment around 2′. This is achieved by Y115, which sits directly under the 2′ site. In this case, the hydrophobic nature of TFV-DP (Figure 5E) and FTC-DP (Figure 5F) interacts well with Y115. Moreover, in the case of FTC-TP, with its unnatural *L*-nucleoside stereochemistry, the inhibitor positions its ring between the primer deoxyribose and Y115. This is not possible for SARS-CoV-2, where the primer ribose 2′OH is not compatible with this binding mode.

## 3. Discussion

The COVID-19 pandemic and initial lack of treatment options fueled scientific innovation and a record-breaking number of reports on potential antivirals for SARS-CoV-2. At the start of the pandemic, it was recognized that nucleotide inhibitors that act on the viral replication machinery were a viable strategy. Remdesivir quickly emerged as an effective agent against SARS-CoV-2 and has been widely used in the treatment of COVID-19 patients. The viral mutagenic nucleotide molnupiravir has also received emergency use authorization for patients with mild to moderate disease. These, and other antiviral nucleotide inhibitors, work by generating a triphosphate metabolite, which is then incorporated into the viral RNA by the RdRp. The inhibitor subsequently disrupts viral replication through several possible mechanisms, including direct chain termination, delayed chain termination, template-dependent chain termination or error catastrophe. An early wave of reports on how NRTIs, used in the treatment of HIV-1, could serve as potent inhibitors to SARS-CoV-2 came as a surprise, being highly inconsistent with the collective knowledge in the nucleoside/tide antiviral discovery field for two reasons. First, HIV-1 RT, an RNA-dependent DNA polymerase which is selective for 2′-deoxynucleotide substrates, is fundamentally different from SARS-CoV-2 RdRp, an RNA-dependent RNA polymerase which is selective for ribose (2′-OH) nucleotide substrates. This essential distinction is well understood by differences in the active sites of the DNA-dependent RNA polymerase and RNA-dependent DNA polymerases, where the RNA polymerases, including SARS-CoV-2′s RdRp, employ a set of polar residues to recognize the ribose 2′-OH, while the DNA polymerases, including HIV-1 RT, rely on more hydrophobic residues to recognize the 2′-deoxyribose. Second, HIV-1 RT, like many viral DNA polymerases, tolerates the removal of 3′-OH on the ribose ring and enables NRTIs to serve as obligate chain-terminators during viral DNA synthesis, as seen in all approved antivirals for DNA viruses and retroviruses, including HSV, HIV, and HBV. On the contrary, the SARS-CoV-2 RdRp appears to have a distinct requirement for an intact 3′-OH ribose, consistent with all approved or published experimental antivirals for RNA viruses [21], such as HCV [22], a panel of respiratory viruses [23], and norovirus [24]. Our study shows that it is highly unlikely that TAF, TDF, or FTC serve as direct antivirals for SARS-CoV-2 for several reasons: (1) the active forms are exceedingly poor substrates for the viral RdRp and highly discriminated against as substrates as shown by both biochemical and structural studies; (2) they showed minimal inhibition of RdRp-catalyzed RNA synthesis, and the inhibition will be further diminished by the high natural NTP cellular concentrations, which are 100–1000-fold higher than natural dNTP; (3) none of the NRTIs tested showed antiviral activity in SARS-CoV-2 infected A549-hACE2 cells or NHBE cultures at concentrations that are >10 or >100-fold higher than the clinical exposure, in the presence of efficient active metabolite formation.

Urgency during a pandemic necessitates evaluation of existing therapies for potential efficacy against a new pathogen. Observational, retrospective, and small nonrandomized trials are sensible to perform in such an environment, and may produce hypotheses worthy of further investigation; however, these studies are more prone to introduction of bias, confounding factors, and chance compared to adequately powered, controlled, randomized clinical trials. The observational study performed by del Amo et al. (2021) demonstrated an association among HIV-positive individuals between treatment with TDF/FTC and lower risk of COVID-19 diagnosis and hospitalization compared to those receiving other ART (medRxiv preprint, https://doi.org/10.1101/2021.11.11.21266189 accessed on 1 June 2022). The proposed rationale for this observation was a potential effect of TDF/FTC against SARS-CoV-2 infection. However, preclinical data presented here disproves this hypothesis, suggesting another mechanism may explain this finding if the association proves true through further clinical evaluation. Strong preclinical data should ideally be available to support a therapy’s mechanism of action prior to inclusion of the agent in interventional randomized controlled trials [25]. For example, preclinical data for remdesivir demonstrating in vitro and in vivo efficacy against coronaviruses Middle East Respiratory Syndrome (MERS-CoV) and SARS-CoV was available prior to inclusion of remdesivir in clinical trials for the novel SARS-CoV-2. This treatment was ultimately shown to be safe and effective in numerous randomized, controlled clinical trials.

In conclusion, the multiple approaches here show that none of the HIV-1 NRTIs tested, including TAF, TDF, TFV, nor FTC, are direct antivirals against SARS-CoV-2. Therefore, these data do not support their use to treat or prevent COVID-19.

## 4. Materials and Methods

### 4.1. Reagents

TAF, TDF, TFV, FTC, RDV, FTC-TP, and RDV-TP are synthesized by Gilead Sciences, Inc. The TFV-DP was from CarboSynth LLC (San Diego, CA, USA).

### 4.2. Antiviral and Cytoxicity Evaluation in A549-hACE2 Cells

A549-hACE2 cells that stably express human angiotensin-converting enzyme 2 (hACE2) were established and provided by the University of Texas Medical Branch [26]. A549-hACE2 cell line was maintained in Dulbecco’s Minimum Essential Medium (DMEM) (Corning, New York, NY, USA) supplemented with 10% fetal bovine serum (FBS) (Hyclone, Logan, UT, USA), 1× Penicillin-Streptomycin-l-Glutamine (Corning, New York, NY, USA) and 10 µg/mL blasticidin (Life Technologies Corporation, Carlsbad, CA, USA). Cells were passaged 2 times per week to maintain sub-confluent densities and were used for experiments at passages 5–20. Compounds were evaluated for SARS-CoV-2 antiviral assay as described previously [27,28]. In brief, A549-hACE2 cells were seeded 1.2 × 10^4^ per well were suspended in 50 μL infection medium and seeded into a white clear-bottom 96-well plate (Corning) and incubated overnight at 37 °C with 5% CO_2_. On the following day, compounds were added directly to cultures as 3-fold serial dilutions with a Tecan D300e digital liquid dispenser, with DMSO volumes normalized to that of the highest compound concentration (final DMSO concentration <1%). SARS-CoV-2-Nluc viruses were diluted to MOI = 0.05 and aliquoted 50 μL/well. At 48 hpi, 75 μL Nano-Glo^®^ luciferase reagent (Promega) was added to each well. Luciferase signals were measured using an Envision microplate reader (Perkin Elmer).

Dose-response curves were fit based on a non-linear regression model. All calculations were performed using GraphPad Prism 8. Data represent the means of two independent experiments consisting of two technical repeats each.

Cytotoxicity was determined in A549-hACE2 cells after 2-day treatment in a 384-well format. Compounds are prepared in 100% DMSO in 384-well polypropylene plates (Greiner, Monroe, NC, USA) with 8 compounds per plate in grouped replicates of 4 at 10 serially diluted concentrations (1:3). The serially diluted compounds were transferred to low dead volume Echo plates (Labcyte, Sunnyvale, CA, USA). The test compounds were spotted to 384-well assay plates (Greiner, Monroe, NC, USA) at 200 nL per well using an Echo acoustic dispenser (Labcyte, Sunnyvale, CA, USA). A549-hACE2 cells were harvested and suspended in DMEM (supplemented with 2% FBS, 1× Penicillin-Streptomycin-l-Glutamine, and 10 µg/mL blasticidin) and seeded to the pre-spotted assay plates at 5000 cells per well for 2-day cytotoxicity assay in 40 µL. The assay plates were incubated for 2 days at 37 °C and 5% CO_2_. At the end of incubation, CellTiter-Glo reagent (Promega, Madison, WI, USA) was prepared. The assay plates and CellTiter-Glo reagent were equilibrated to room temperature for at least 15 min. An amount of 40 µL per well of CellTiter-Glo reagent was added and mixed 5 times with a Biomek FX (Beckman Coulter Life Sciences, Indianapolis, IN, USA) before reading the luminescence signal on an EnVision multimode plate reader (Perkin Elmer, Waltham, MA, USA). Fingolimod was used as positive control and DMSO was used as negative control. Values were normalized to the positive and negative controls (as 100% and 0% cytotoxicity, respectively) and data was fitted using four-parameter non-linear regression analysis. The CC_50_ value for each compound was defined as the concentration reducing the viability by 50%.

### 4.3. Evaluation of Antiviral Assays in NHBE Cells

Normal human bronchial epithelial (NHBE) cultures from donor 41219 were purchased from Lonza (Walkersville, MD, USA) and maintained in bronchial epithelial cell growth medium (BEGM) (Lonza, Walkersville, MD, USA) with all provided supplements in the BulletKit. Cells were passaged 2–3 times per week to maintain sub-confluent densities and were used for experiments at passages 2–4. Compounds were evaluated for SARS-CoV-2 antiviral assay as described previously [29]. Briefly, NHBE cells were seeded in 24-well plates at 1 × 10^5^ cells per well at a final volume of 500 μL BEGM^TM^. Cultures were incubated overnight at 37 °C with 5% CO_2_. On the following day, media was replaced with 500 μL growth medium. Cultures were treated with 1:3 serial dilutions of compound using the HP D300e digital dispenser with normalization to the highest concentration of DMSO in all wells <1% final volume. The cells were then infected with 100 μL SARS-CoV-2-Fluc diluted in BEGM media at MOI = 5. Uninfected and untreated wells were included as controls to determine compound efficacy against SARS-CoV-2-Fluc. Following incubation with compound and virus for 24 h at 37 °C with 5% CO_2_, culture supernatant was removed from each well and replaced with 300 μL of ONE-Glo™ luciferase reagent (Promega). The plates were shaken at 400 rpm for 10 min at room temperature, 200 μL of supernatant from each well was transferred to a 96-well opaque plate (Corning) and luminescence signal was measured using an Envision microplate reader (PerkinElmer). Values were normalized to the uninfected and infected DMSO controls (0% and 100% infection, respectively).

### 4.4. Measurement of Active Metabolites in Cells

Concentrations of intracellular metabolites following incubation of compounds were determined as described by Li, et al. [12]. Briefly, following cell isolation and wash with cold 1 × Tris-HCl (pH 7.0)-buffered saline, each A549 (CCL-185, ATCC; Manassas, VA, USA) cell sample was then treated with 500 μL of dry ice-cold extraction buffer (0.1% potassium hydroxide and 67 mM ethylenediamine tetra-acetic acid in 70% methanol, containing 0.5 μM chloro-ATP as internal standard). The above solution was vortexed for 15 min and then centrifuged at 3000× *g* rpm for 15 min. The supernatant was transferred to a clean 96-deep-well plate and then dried on an evaporator. Once dry, samples were reconstituted with 80 μL of 1 mM ammonium phosphate buffer (pH 7) and transferred to a 96-short-well plate for analysis. An aliquot of 10 μL was injected into a Sciex QTrap 6500 + liquid chromatography-tandem mass spectrometry (LC-MS/MS) system run under the multiple reaction monitoring (MRM) operation mode.

Standard calibration curves were constructed based on pmol of compound per sample. The standard curve was prepared by spiking an appropriate amount of FTC and its monophosphate and diphosphate metabolites, FTC 5′-monophosphate (FTC-MP) and FTC 5′-diphosphate (FTC-DP) solutions prepared in water, into untreated A549 matrix and then serially diluted to complete the calibration standards. Metabolite concentrations were quantified in each sample and the intracellular concentrations (pmol/million cells) were determined by dividing the value by the cell number in the sample.

### 4.5. Biochemical Assays

#### 4.5.1. SARS-CoV-2 RdRp Expression and Purification

The pFastBac-1 (Invitrogen, Burlington, ON, Canada) plasmid with the codon-optimized synthetic DNA sequences (GenScript, Piscataway, NJ, USA) coding for SARS-CoV-2 (NCBI: QHD43415.1) nsp5, nsp7, nsp8, and nsp12 were used as a starting material for protein expression in insect cells (Sf9, Invitrogen, Waltham, MA, USA) [14,15]. We employed the MultiBac (Geneva Biotech, Indianapolis, IN, USA) system for protein expression in insect cells (Sf9, Invitrogen) according to published protocols [30,31]. SARS-CoV-2 protein complexes were purified using nickel-nitrilotriacetic acid affinity chromatography of the nsp8 *N*-terminal 8-histidine tag according to the manufacturer’s specifications (Thermo Fisher Scientific, Waltham, MA, USA).

#### 4.5.2. Selective Incorporation of Nucleotide Analogs

RNA synthesis was monitored with short model RNA primer/template substrates mimicking elongation complexes. The following synthetic 5′-monophosphorylated RNA templates were used in this study (the portion of the template which is complementary to the 4-nt primer is underlined): 3′-UGCGCUAGAAAAAAp for measurements of TFV-DP, and 3′-UGCGCGAUAAAAAAp for measurements of FTC-TP single nucleotide incorporation kinetics parameters V_max_ and *K*_m._ The ratio of V_max_ over *K*_m_ provides a measure for the efficiency for nucleotide incorporation. The enzyme concentration is the same in both cases and therefore cancels out during the calculation of relative efficiencies. The relative efficiency values are expressed as fold difference from the efficiency of incorporation of ATP (RDV-TP and TFV-DP) and CTP (for FTC-TP).

Briefly, the selective incorporation assay consisted of mixing (final concentrations) Tris-HCl (pH 8.0, 25 mM), RNA primer (200 μM), RNA template (2 μM), [α-^32^P] GTP (0.1 μM), and various concentrations and combinations of NTP or dNTP analogs were optimized to avoid misincorporation at subsequent positions. The concentration of the SARS-CoV-2 RdRp complex chosen (150 nM) ensured the incorporation of [α-^32^P] GTP was linear. Reactions were stopped after 30 min by the addition of equal volume formamide/EDTA (50 mM) and incubated at 95 °C for 5 min. The reaction products were resolved through 20% PAGE, and the [α-^32^P]-generated signal was stored and scanned from phosphorimager screens. Data were collected and analyzed using GraphPad Prism 7.0 (GraphPad Software, Inc., San Diego, CA, USA) as reported previously [14,15,32,33,34,35]. This assay involves incorporation of [α-^32^P] GTP into the 4-nt-primer. With this approach, it is not possible to quantify the fraction of the 4-nt primer that has been extended by [α-^32^P] GMP; however, the reaction products are clearly defined by the correct incorporation of [α-^32^P] GTP.

#### 4.5.3. Inhibition of SARS-CoV-2 RdRp-Catalyzed RNA Synthesis by Nucleotide Analogs in a Gel-Based Assay

Using the same RNA primer/template substrates mentioned above and an NTP cocktail containing ATP, CTP, and UTP (1 µM), inhibition of RNA synthesis by each respective nucleotide analog was monitored in a dose-dependent manner. The assay was performed and visualized as described above, differing in the supplementation of an NTP cocktail to support the synthesis of full template-length product.

#### 4.5.4. Inhibition of SARS-CoV-2 RdRp in a Filter-Binding Assay

Filter binding assays to measure inhibition of SARS-CoV-2 RdRp were modified from previous studies [36]. Enzyme reactions were initiated by mixing 3× mixtures of enzyme, inhibitors, and NTPs in equal parts. The enzyme mixture contained final reaction concentrations of 25 mM HEPES pH 7.5, 10 mM KCl, 50 mM NaCl, 0.1 μM recombinant nsp12, 0.3 μM nsp7 and nsp8, and 0.5 μM of hybridized dsRNA (R30/67; 30-mer: 5′-AGGGAAGGAGGAAGAAAGAAAGGAGGGAGG-3′; 67-mer: 3′-UCCCUUCCUCCUUCUUUCUUUCCUCCCUCCGGCCCGCCGUAGCCCCCAGCUUCAGUAGGCUACAGUC-5′, underlined region is where ‘30-mer hybridizes to 67-mer). The NTP mixture varied depending on the inhibitor studied. For RDV-TP and TFV-DP studies, ATP is the competing nucleotide and for FTC-TP studies CTP is competing. The mixture contained final reaction concentrations of 2 mM MgCl_2_, 0.1 μCi/μL [α-^32^P] ATP or [α-^32^P] CTP, 1 μM ATP or CTP, and 10 μM non-competing natural NTPs. The inhibitors were serially diluted three-fold so that final reaction concentrations ranged from 10 μM to 13.7 nM. After mixing, reactions proceeded for 20 min at 37 °C before spotting 5 μL of the reaction onto DE81 filter paper. The paper was air dried, washed with 0.125 M Na_2_HPO_4_ three times for five minutes, water for five minutes, and ethanol for an additional five minutes. The paper was then air dried and exposed to a Typhoon phosphor imager screen. The screen was scanned by an Amersham Typhoon Imager (GE Healthcare, Piscataway, NJ, USA) and spots were quantified using ImageQuant. IC_50_ values were calculated by dividing the signals by a no-inhibitor control to obtain a percent activity value, and then plotted against concentration in GraphPad Prism. The data was fit to a four-parameter sigmoidal dose response with variable slope.

### 4.6. Structural Aanalysis

Modeling of the critical nucleoside triphosphate pre-incorporation complex of the SARS-CoV-2 RdRp followed work described previously [14,15,17]. Briefly, two Mg^++^ ions and a nucleoside triphosphate or inhibitor analog were modeled into the active site of the RNA duplex-bound RdRp complex, composed of nsp12, nsp7, two subunits of nsp8, and two subunits of nsp13 (pdb 6XEZ) [37]. Structures were initially refined using standard protein preparation procedures in the Schrödinger software package. Active site sidechain conformations were altered to be consistent with substrate and metal binding based on existing ternary structures of other RdRps [18]. Specifically, D618, D760, and D761 coordinate the two Mg^++^ ions, which in turn coordinate the triphosphate of the substrate. The triphosphate is further stabilized by the basic residues R555 and K798. D623 and S682 shift to recognize the substrate 2′OH when it is present, while the nucleotide base Watson-Crick pairs with the appropriate template. These substrate-bound models were further minimized using the OPLS4 forcefield in Macromodel (Schrödinger Release 2022–1: Macromodel, Schrödinger, LLC., New York, NY, USA). Initial binding modes for the inhibitors followed their binding modes in HIV-RT. For HIV-RT, existing ternary structures of pre-incorporated TFV-DP (pdb 1T05) [38], FTC-DP (pdb 6UJX) [39], dATP (pdb 3KK2) [40], and dCTP (pdb 6P1I) [41] were modified to incorporate a missing 3′OH on the primer nucleotide and to add a missing catalytic Mg^++^ ion. Protein refinement followed a similar procedure to that of the SARS-CoV-2 RdRp.

## Figures and Tables

**Figure 1 molecules-27-04212-f001:**
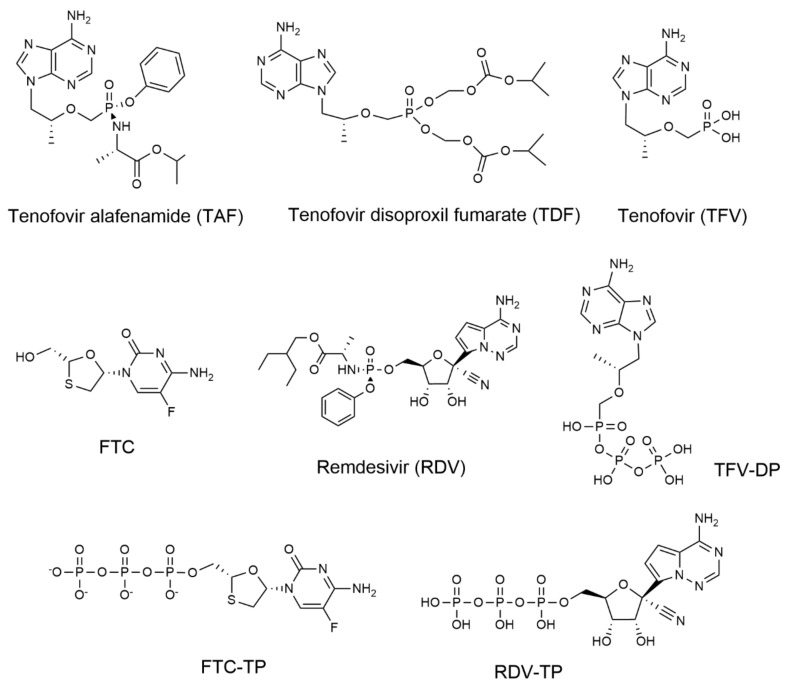
Chemical structures of nucleoside/tide analogs used in this study. TAF, TDF, TFV, FTC, and RDV are used in the cell-based assays, while the active metabolites TFV-DP, FTC-TP, and RDV-TP are evaluated in the biochemical studies.

**Figure 2 molecules-27-04212-f002:**
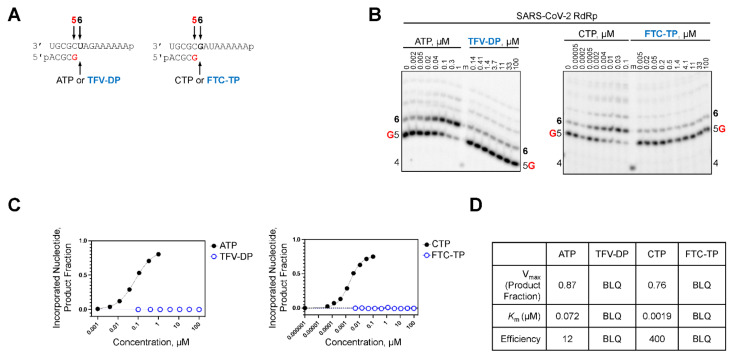
TFV-DP and FTC-TP are poor substrates for SARS-CoV-2 RdRp. (**A**) RNA primer/templates used to measure the incorporation of TFV-DP and FTC-TP by SARS-CoV-2 RdRp; the sequences support the single incorporation of ATP (**left**) and CTP (**right**) analogs at position six; G indicates the incorporation of [α-^32^P] at position five (red). (**B**) migration pattern of the products of RNA synthesis catalyzed by SARS-CoV-2 RdRp. A 5′-^32^P-labeled 4-nt primer (4) serves as a size marker (m). (**C**) graphical representation of the data shown in B, fitting the data points to Michaelis-Menten function to determine V*_max_* and *K*_m_ shown in D and Table 3. (**D**) kinetic parameters and efficiency of TFV-DP and FTC-TP incorporation by SARS-CoV-2 RdRp. Efficiency is determined as the quotient of V_max_ over *K*_m_, the signal generated as a result of TFV-DP and FTC-TP incorporation was below the level of quantification (BLQ). Efficiency is determined as the quotient of V_max_ over *K*_m_.

**Figure 3 molecules-27-04212-f003:**
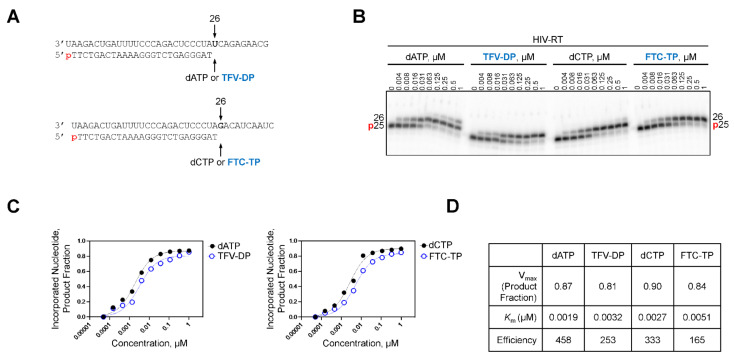
TFV-DP and FTC-TP are good substrates for HIV-1 RT. (**A**) radiolabeled DNA primer/templates used to measure the incorporation of TFV-DP and FTC-TP by HIV-1 RT, the sequences support the single incorporation of ATP (**top**) and CTP (**bottom**) analogs at position 26, p indicates the position of [α-^32^P] to label the primer for visualization (red). (**B**) migration pattern of the products of DNA synthesis catalyzed by HIV-RT. (**C**) graphical representation of the data shown in (**B**), fitting the data points to Michaelis-Menten function to determine V_max_ and *K*_m_ shown in (**D**) and Table 3. (**D**) kinetic parameters and efficiency of TFV-DP and FTC-TP incorporation by HIV-RT. Efficiency is determined as the quotient of V_max_ over *K*_m_.

**Figure 4 molecules-27-04212-f004:**
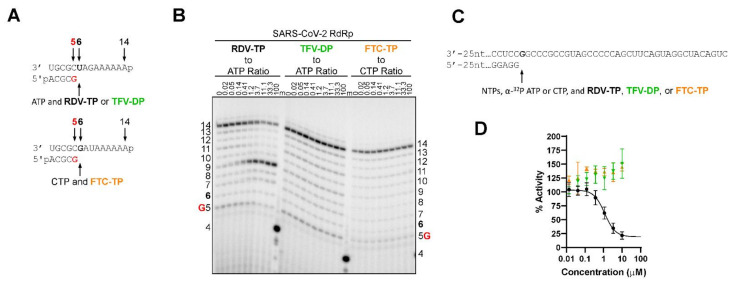
TFV-DP and FTC-TP do not inhibit SARS-CoV-2 RdRp-catalyzed RNA synthesis. (**A**) RNA primer/templates used to monitor the incorporation of RDV-TP, TFV-DP, and FTC-TP and subsequent RNA synthesis catalyzed by SARS-CoV-2 RdRp in the presence of constant NTP concentration; the sequences support the single incorporation of ATP (**top**) and CTP (**bottom**) analogs at position six; G indicates the incorporation of [α-^32^P] at position five (red). (**B**) migration pattern of the products of RNA synthesis catalyzed by SARS-CoV-2 RdRp. A 5′-^32^P-labeled 4-nt primer (4) serves as a size marker (m). (**C**) RNA primer/template used to monitor the inhibition of NTP incorporation catalyzed by SARS-CoV-2 RdRp in the presence of RDV-TP, TFV-DP, or FTC-TP; enzyme activity was detected by incorporation of radio-labelled nucleotides. Sequence depicted shows the partial RNA/RNA 30-mer primer/67-mer template where NTP incorporation begins (**bolded G**). (**D**) dose-response curves of TFV-DP (green), FTC-TP (orange), and RDV-TP (black) in a filter-based SARS-CoV-2 RdRp biochemical assay.

**Figure 5 molecules-27-04212-f005:**
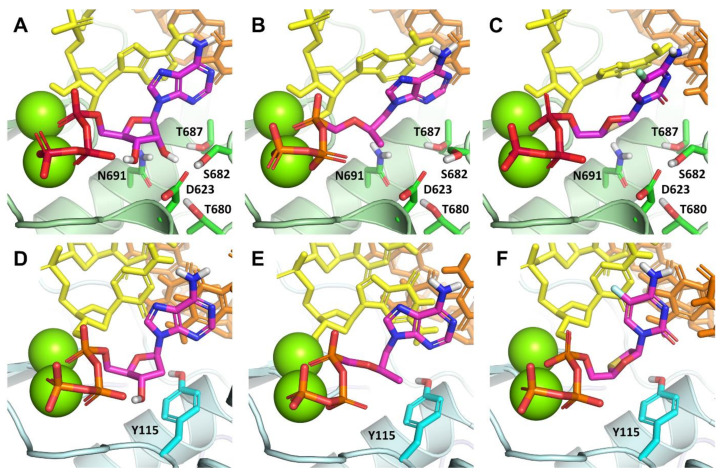
Models of (**A**) ATP, (**B**) TFV-DP and (**C**) FTC-TP in the pre-incorporation state of the SARS-CoV-2 nsp12 RdRp. The RdRp features a set of polar residues (D623, T680, S682, T687, and N691) which form a hydrogen bond network to discriminate between ribose substrates and 2′-deoxyribose substrates. TFV-DP and FTC-TP both lack the capacity to interact with these polar residues, instead presenting an incompatible hydrophobic face toward these residues. In addition, the unnatural *L*-nucleoside stereochemistry of FTC-TP clashes with the primer 2′-OH, leading to further distortion. In contrast, HIV-RT features a hydrophobic environment at 2′ from Y115. This creates an ideal situation for coordination of (**D**) dATP, (**E**) TFV-DP, and (**F**) FTC-TP, discriminating instead against ribose substrates.

**Table 1 molecules-27-04212-t001:** TAF, TDF, TFV, and FTC are not active against SARS-CoV-2 in cell-based assays.

Compounds	Cell-Based Antiviral Assay EC_50_ (μM)		Clinical Drug Exposure and Active Metabolite Levels in PBMC	
Nucleoside/Tide Analogs	A549-hACE2 NanoLuc Reporter ^1^	NHBE Firefly Luc Reporter ^2^	Drug Exposure (μM) (Active Metabolite) ^3^	Plasma Protein Binding (%)
RDV	0.104 ± 0.016	0.037 ± 0.017	C_max_ = 7.3, 3.7 ** (10.2 μM in PBMC)	88
TAF	>50	>50	C_max_ = 0.4 μM	80
TDF	>50	>50	Not detectable	-
TFV	>500	Not done	NA ^4^	-
FTC	>50	>50	C_max_ = 7.9 μM	4

^1^ Values are mean ± standard deviation of two to four independent replicates in A549-hACE2 cells (human alveolar epithelial cell line expressing human ACE2). TAF, TDF, and FTC showed no cytotoxicity up to 50 μM after a 2-day incubation. TFV showed no cytotoxicity up to 500 μM after a 2-day incubation. RDV CC_50_ ≥ 16.7 μM in the 2-day assay. ^2^ Values are mean ± standard deviation of three independent replicates in NHBE cells (normal human bronchial epithelial cells). ^3^ References [8,9]. Values represent C_max_ and C_min_ for human exposures; ** 200/100 mg QD IV regimen. ^4^ NA = not applicable.

**Table 2 molecules-27-04212-t002:** TAF, TDF, and FTC formed high levels of active metabolites in cells that are significantly higher than the clinically-relevant concentrations at the currently approved doses for the treatment of HIV-1.

Nucleoside/Tide Analogs	Active Metabolite Formed in Cell Culture (μM) in A549 at 24-h	Extrapolated Active Metabolite under EC_50_ (μM) ^1^ in A549 at 24-h	Metabolite Level under Clinical Exposure (μM) ^2^ Post Multiple Doses
RDV	21.0 ± 2.2 (with 1 μM RDV)	1.4	10.2 (with RDV 200/100 mg QD)
TAF	126 ± 50 (with 1 μM TAF)	>6300	1 (with TAF 25 mg QD)
TDF	14 ± 2 (with 1 μM TDV)	>700	0.2 (with TDF 300 mg QD)
TFV	1.7 ± 0.6 (with 10 μM TFV)	NA^3^	NA ^3^
FTC	5.8 ± 1.8 (with 1 μM FTC)	>290	20 (with FTC 200 mg QD)

^1^ Calculated from the active metabolite levels measured in the A549 cells treated with 1 or 10 μM of drugs and the EC_50_ values reported in Table 1. A549 cell volume was reported as 1.67 pL by Jiang et al. [13]. ^2^ References [8,9]. ^3^ NA = Not applicable.

**Table 3 molecules-27-04212-t003:** Active metabolites TFV-DP and FTC-TP are very poor substrates for SARS-CoV-2 RdRp while remaining to be good substrates for HIV-1 RT in biochemical assays.

Nucleoside Analogs or Nucleotide Prodrugs	NTP or Active Metabolites	SARS-CoV-2 RNA Polymerase Assay ^1^			
		V_max_ (Product Fraction) ^2^	*K*_m_ (μM)	Incorporation Efficiency (V_max_/*K*_m_)	Efficiency Relative to the Natural NTP Substrate
Adenosine	ATP	0.87	0.072	12	-
RDV	GS-443902	0.74	0.0089	83	3.6-fold higher than ATP ^3^
TAF, TDF, TFV	TFV-DP	BLQ ^4^	BLQ ^4^	BLQ ^4^	Too low to be measured
Cytidine	CTP	0.76	0.019	400	-
FTC	FTC-TP	BLQ ^4^	BLQ ^4^	BLQ ^4^	Too low to be measured
**Nucleoside Analogs or Nucleotide Prodrugs**	**NTP or** **Active** **Metabolites**	**HIV-1 RT DNA Polymerase Assay ^1^**			
		**V_max_**	***K*_m_ (μM)**	**Incorporation Efficiency** **(V_max_/*K*_m_)**	**Efficiency** **Relative to the Natural dNTP Substrate**
2′-deoxyadenosine	dATP	0.87	0.0019	458	-
TAF, TDF, TFV	TFV-DP	0.81	0.0032	253	1.5-fold lower than dATP
2′-deoxycytidine	dCTP	0.90	0.0027	333	-
FTC	FTC-TP	0.84	0.0051	165	2-fold lower than dCTP

^1^ Incorporation efficiency of the 5′-triphosphate form of each drug was evaluated against the SARS-CoV-2 RNA polymerase HIV-1 RT DNA polymerase. The analog incorporation efficiency is expressed as the quotient of the Michaelis-Menten parameters V_max_/*K*_m_, while comparison of efficiencies is expressed as fold difference from the efficiency of incorporation of ATP (for RDV, TAF, and TDF) and CTP (for FTC). ^2^ Product fraction is the measure of nucleotide or nucleotide analog incorporation at position six, the signal is generated due to extension of [α-^32^P] GTP incorporated immediately before at position five. The value is determined by dividing the signal of position six by the total signal in the lane (position five and six). ^3^ Data and fold difference correspond to Gordon et al., (2020) [14]. ^4^ BLQ = below level of quantification.

**Table 4 molecules-27-04212-t004:** Active metabolites TFV-DP and FTC-TP are poor inhibitors for SARS-CoV-2 RdRp-catalyzed RNA synthesis.

Nucleoside Analogs or Nucleotide Prodrugs	Active Metabolites	Inhibition of SARS-CoV-2 RdRp IC_50_ (μM) ^1^
TAF, TDF, TFV	TFV-DP	>100
FTC	FTC-TP	>100
RDV (positive control)	RDV-TP	1.0 ± 0.3

^1^ The reported values are the average ± standard deviation of three independent measurements.

## Data Availability

Not applicable.

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
