# Peer review of "The Nucleoside/Nucleotide Analogs Tenofovir and Emtricitabine Are Inactive against SARS-CoV-2"

_molecules, 2022, doi:10.3390/molecules27134212_

Round 1

Reviewer 1 Report

The manuscript entitled: The Nucleoside/nucleotide Analogs Tenofovir and Emtricitabine are Inactive Against SARS-CoV-2 documents the effectiveness of the approved HIV-1 NRTIs  TFV, FTC, TAF, and TDF for their antiviral effect against SARS-CoV-2. Authors have performed several in vitro and in silico assays and have demonstrated that these approved anti-HIV-1 drug candidates are ineffective against SARS-CoV-2. The results are convincing and essential for the scientific community and clinicians to develop therapeutic approaches against COVID-19. However, there are a few concerns mentioned below.

·        All the in vitro and in silico should be written in italics throughout the manuscript. And all the EC50 should be subscripted and presented as EC50

·        Line #45 PLWH full form?

·        Line #51 PLWHA full form? Also, is it the same as PLWH?

·        The units of the drugs are confusing in certain places. For example, on lines #77 and #78, is it µM or M? Same is the case in the abstract. All the units should be uniform throughout the manuscript.

·        A figure including the chemical structures will be beneficial with reference to docking studies. Please include them either in the main text or as a supplementary figure.

·        The abstract mentions, “when exploring ideas of using these and other NRTIs to treat or prevent COVID-19” However, the other NRTIs are not discussed. Will the other NRTIs and NNRTIs demonstrate the same efficacy? Adding some discussion will be helpful.

·        Will the drugs also be inactive against other variants of SARS-CoV-2? Considering the mutations observed in the variants, will the drug show different activity profile? Adding a sentence or two in the discussion on this will be helpful.

·        Figure #3, Use better resolution images, especially for panel B. 

Reviewer 2 Report

They are included in the enclosed attachment

Author Response

Reviewer #2: I have analysed the manuscript titled “The Nucleoside/nucleotide Analogs Tenofovir and Emtricitabine are Inactive Against SARS-CoV-2”. The development of antiviral agents for Covid-19 is of great interest today. The manuscript is well written and easy to read. The authors indicate using in vitro data that TFV, TAF, TDF and FTC nucleoside/tides are inactive against SARS-CoV-2. The cells used by the authors seem interesting, but there are other cells. I think that the manuscript may favour that other authors study these antiviral drugs in other cells to confirm it. Only few minor points can be pointed, which are as follows.

Response: The reviwer is correct that there are other cells we could study the antiviviral effect in; however, we believe the lung cells we used are most relevant to COVID-19 infection in human.

Minor points:

  1. Figs 1-3 appear with low resolution and most values are too small to see. The legend of these figures is too long and it can be included within the text. Some additional explanation is necessary to understand the meaning of some inserted figures and tables.

Response: We fully agree with the reviewer’s suggestions. We lost image resolution when the system compiled our submitted files. We will submit high resolution image file for the final submission. We have also revised the Result section to provide more explaination on the inserted figures and tables.

  1. What MM or DFT theoretical methods have been used in the software package to optimize the structures shown in Fig. 4.

Response: We want to thank the reviewer for this feedback and we have revised the manuscript to include more detailed information on the Methods under Section 4.6.

Reviewer 3 Report

The study «The Nucleoside/nucleotide Analogs Tenofovir and Emtricitabine are Inactive Against SARS-CoV-2» is performed at a good technical level and the results are well presented.

 For a better representation of the experimental work, I would recommend:

1) Giving the structural formulas of all nucleosides/nucleotides whose activity has been studied.

2) Too many abbreviations in the text. It is advisable to provide a complete list of them before the References.

3) There are a number of comments:

 Line 7             2 Department of Medical Microbiology and Immunology ‒ slide to Line 8

 Line 16           «set of in vitro...  ‒ in vitro - italic

Line 18           «50  M TAF, TDF, FTC». Is that µM? Same for page 1 line 19, page 2 lines 77 and 78.

Line 29           «asrecently as...» ‒ as recently as

From 29 line to the end: references must be in [1] not (1).

Line 41           in silico must be  italic

Line 47           «other therapies.(6).» ‒ other therapies [6].

Line 51           PLWHA = PLWH?

Line 77           EC50 = EC50 ‒ subscript?

4) Information about the units of Vmax is missing (Table 3).

5) The captions of Figures 1 and 2 indicate "… to determine Vmax and Km shown in D and Table 2." but the Vmax and Km are shown in Table 3.

 6) On Figure 4 the results of nucleotide docking into the active sites of two enzymes are presented. It is desirable to improve the resolution of the drawings, because when they are enlarged, everything blurs. There is no description of why the nucleotides in active sites are positioned in this way, hydrogen bonds with amino acids of active centers are not shown (neither for ATP, nor for the studied nucleotides). It is not clear what kind of nucleotides  (yellow structures) are also located in the active site. The interaction of heterocyclic nucleotide residues with hydrophobic amino acids of active sites is not shown. As presented, the docking data is not informative for readers.

7) Submit a list of references as required by the journal:

1. Author 1, A.B.; Author 2, C.D. Title of the article. Abbreviated Journal Name Year, Volume, page range.

Author Response

Reviewer #3: The study «The Nucleoside/nucleotide Analogs Tenofovir and Emtricitabine are Inactive Against SARS-CoV-2» is performed at a good technical level and the results are well presented.

For a better representation of the experimental work, I would recommend:

  • Giving the structural formulas of all nucleosides/nucleotides whose activity has been studied. Too many abbreviations in the text. It is advisable to provide a complete list of them before the References.

Response: We have added Figure 1 to show all chemcial structures and added a summary list of all abbreviations before the References section.

  • There are a number of comments:

Line 7             2 Department of Medical Microbiology and Immunology ‒ slide to Line 8

Line 16           «set of in vitro...  â€’ in vitro - italic

Line 18           «50  M TAF, TDF, FTC». Is that µM? Same for page 1 line 19, page 2 lines 77 and 78.

Line 29           «asrecently as...» ‒ as recently as

From 29 line to the end: references must be in [1] not (1).

Line 41           in silico must be  italic

Line 47           «other therapies.(6).» ‒ other therapies [6].

Line 51           PLWHA = PLWH?

Line 77           EC50 = EC50 â€’ subscript?

Response: We thank the reviewer to point out these typoes which are derived from two source: (1) the misspellings such as “asrecently, in silico, or PLWHA” are made by us and they are all corrected in the revision; (2) the mistranformation from our initial word file to the printset version such as the misrepresentation or misplacement of mM and EC50 need to be corrected by the editors.

  • Information about the units of Vmax is missing (Table 3).

Response: We thanks the reviewer for pointing out this oversight from us. We have added more information in Table 3.

4) The captions of Figures 1 and 2 indicate "… to determine Vmax and Km shown in D and Table 2." but the Vmax and Km are shown in Table 3.

Response: We thank the reviwer for identifying this typo. We have corrected it in the revision.

5) On Figure 4 the results of nucleotide docking into the active sites of two enzymes are presented. It is desirable to improve the resolution of the drawings, because when they are enlarged, everything blurs. There is no description of why the nucleotides in active sites are positioned in this way, hydrogen bonds with amino acids of active centers are not shown (neither for ATP, nor for the studied nucleotides). It is not clear what kind of nucleotides  (yellow structures) are also located in the active site. The interaction of heterocyclic nucleotide residues with hydrophobic amino acids of active sites is not shown. As presented, the docking data is not informative for readers.

Response: Please see updated version in the Method section under Section 4.6 addressing the reviewer’s comment. We have also uploaded high-resolution image files for the final submission.  

6) Submit a list of references as required by the journal:

  1. Author 1, A.B.; Author 2, C.D. Title of the article. Abbreviated Journal NameYearVolume, page range.

Response: We thank the viewer for this suggestion, and we have updated the reference style following the journal's instructions.